# A new automatic sugarcane seed cutting machine based on internet of things technology and RGB color sensor

Liu Yang[1], Loai S. Nasrat[2], Mohamed E. Badawy[3], Daniel Eutyche Mbadjoun Wapet[4]*, Manar A. Ourapi[5], Tamer M. El-Messery[6], Irina Aleksandrova[6], Mohamed Metwally Mahmoud[7], Mahmoud M. Hussein[7,8], Abdallah E. Elwakeel[5]

1 School of Mechanical and Electrical Engineering, Shihezi University, Xinjiang, China, 2 Electrical Power Engineering Department, Faculty of Engineering, Aswan University, Aswan, Egypt, 3 Agricultural Engineering Research Institute, Giza, Egypt, 4 National Advanced School of Engineering, Université de Yaoundé I, Yaoundé, Cameroon, 5 Agricultural Engineering Department, Faculty of Agriculture and Natural Resources, Aswan University, Aswan, Egypt, 6 International Research Centre "Biotechnologies of the Third Millennium", Faculty of Biotechnologies (BioTech), ITMO University, St. Petersburg, Russia, 7 Electrical Engineering Department, Faculty of Energy Engineering, Aswan University, Aswan, Egypt, 8 Department of Communications Technology Engineering, Technical College, Imam Ja'afar Al-Sadiq University, Baghdad, Iraq

* eutychedan@gmail.com

**Data Availability Statement:** All relevant data are within the manuscript.

## Abstract

Egypt is among the world's largest producers of sugarcane. This crop is of great economic importance in the country, as it serves as a primary source of sugar, a vital strategic material. The pre-cutting planting mode is the most used technique for cultivating sugarcane in Egypt. However, this method is plagued by several issues that adversely affect the quality of the crop. A proposed solution to these problems is the implementation of a sugarcane-seed-cutting device, which incorporates automatic identification technology for optimal efficiency. The aim is to enhance the cutting quality and efficiency of the pre-cutting planting mode of sugarcane. The developed machine consists of a feeding system, a node scanning and detection system, a node cutting system, a sugarcane seed counting and monitoring system, and a control system. The current research aims to study the pulse widths (PW) of three-color channels (R, G, and B) of the RGB color sensors under laboratory conditions. The output PW of red, green, and blue channel values were recorded at three color types for hand-colored nodes [black, red, and blue], three speeds of the feeding system [7.5 m/min, 5 m/min, and 4.3 m/min], three installing heights of the RGB color sensors [2.0 cm, 3.0 cm, and 4.0 cm], and three widths of the colored line [10.0 mm, 7.0 mm, and 3.0 mm]. The laboratory test results s to identify hand-colored sugarcane nodes showed that the recognition rate ranged from 95% to 100% and the average scanning time ranged from 1.0 s to 1.75 s. The capacity of the developed machine ranged up to 1200 seeds per hour. The highest performance of the developed machine was 100% when using hand-colored sugarcane stalks with a 10 mm blue color line and installing the RGB color sensor at 2.0 cm in height, as well as increasing the speed of the feeding system to 7.5 m/min. The use of IoT and RGB color sensors has made it possible to get analytical indicators like those achieved with other

**Funding:** The author(s) received no specific funding for this work.

**Competing interests:** The authors have declared that no competing interests exist.

**Abbreviations:** HSV, Herpes simplex virus; ASSCM, automatic sugarcane seed cutting machine; DC, Direct current; IoT, Internet of things; PW, pulse widths; RGB, Red, green, and blue; TRD, time, rate, and distance; ms, Millisecond; Nomenclature $F_t$, tangential impact of circular saw blades on the cutting area, N; $F_{N1}$, Supporting force of side baffle on sugarcane, N; $\theta$, Angle at which the blade cuts the sugarcane; $C$, The speed of the ultrasonic waves, m/s; $d$, Distance, m; H, Height of sugarcane seeds guiding tube, m; D, Diameter of the guiding tube, m; $\xi$, The path returned by the disk knife; $\psi$, Vertically upward direction; $G$, Sugarcane gravity; $T$, Time value from transmitter to sugarcane seeds back to receiver, s; $F_n$, The applied pressure due to the circular saw blades at the cutting site, N; $F_{N2}$, The influence of the magnitude—Strength of the soil support on sugar cane, N.

automatic systems for cutting sugar cane seeds without requiring the use of computers or expensive, fast industrial cameras for image processing.

## 1. Introduction

Sugar is an important agricultural product in Egypt, and sugarcane is the main sugar crop. The sugarcane industry provides support for economic development and the increment of farmers' income. Mechanization and refinement of the whole sugarcane planting process is a trend of industrial development, but most sugarcane seed cutting machines in the world do not have an anti-injury function to prevent damage to buds during the automatic sugarcane seed cutting process, which restricts the development of the sugarcane industry [1–4]. Identifying the stem nodes in sugarcane is a crucial technology for advancing the intelligence and mechanization of the sugarcane industry. Nevertheless, the rapid and precise detection of these stem nodes continues to pose a significant obstacle [5, 6].

Most planting machines used worldwide lack the capability to prevent bud damage during the automatic cutting process of sugarcane seeds. Real-time automatic planters typically rely on fixed-length cutting, which offers higher efficiency compared to manual planting [7–10]. However, this approach often results in a high rate of bud injury. Pre-cutting planters, on the other hand, utilize pre-cut seeds to avoid damaging the buds and achieve greater efficiency than real-time cutting machines [11, 12]. Nonetheless, the existing sugarcane cutting machines are unable to automatically locate the cane buds, leading to lower cutting efficiency [13–15]. To enhance the quality of sugarcane seeds and improve cutting efficiency, it is necessary to develop seed cutting technology and equipment that can effectively prevent bud damage.

Previous studies have demonstrated the effectiveness of various technologies, such as machine vision, machine learning, deep learning, wavelet analysis, image processing algorithms, and the Herpes simplex virus (HSV) color space in identifying sugarcane stem nodes and avoiding them during seed cutting [16–25]. Wang et al. [26] developed and later tested a system for cutting sugarcane seeds using machine vision in the seed pretreatment mode. The results showed that the recognition rate of cut sugarcane seeds was at least 94.3%, and the accuracy level ranged from 94.3% to 98%, and the average accuracy was 98.2%. This result allowed us to find that the level of injury to buds does not exceed 3.8%, and the average time spent on cutting one seed is 0.7 seconds. This demonstrates that the cutting system has achieved high cutting and recognition rates while maintaining a low injury rate. In a similar vein, Zhou et al. [5] developed an identification and localization algorithm for sugarcane stem nodes by combining YOLOv3, a popular object detection algorithm, with traditional computer vision methods. The results of the experiments conducted in this study indicate that the stem node recognition algorithm achieved high precision, recall, and harmonic mean rates, 99.68%, 100%, and 99.84%, respectively. When compared to the YOLOv3 network, the algorithm demonstrated improvements of 2.28% in precision rate and 1.13% in harmonic mean. Zhou et al. [21] introduced a new design of sugarcane seed cutting systems based on machine vision. The offline identification of sugarcane stalk segments yielded a recognition rate of 93% with an average processing time of 0.539 seconds. With a single cutting unit, the developed system has a throughput capacity that can reach up to 2400 buds per hour. During the online test, it was observed that the cutting point achieved a satisfactory level of position precision, meeting the requirements of agricultural operations. Additionally, there was no incidence of bud damage, indicating a high level of precision and accuracy in the cutting process. Moshashai et al. [16]

conducted a preliminary study on sugarcane node identification using gray image threshold segmentation. Meng et al. [27] explored sugarcane node recognition technology based on wavelet analysis. Chen et al. [23] investigated a wavelet-based approach for recognizing sugarcane stem nodes. The study reported a standard deviation of 0.494 mm and a maximum value of 9.99 mm for the detected nodes. The approach achieved a detection rate of 99.63% for cane seed samples, with an error rate of 0.37% and a response time of 0.25 seconds. In another work by Chen et al. [19] an object detection algorithm based on deep learning was proposed for the recognition of sugarcane stem nodes in complex natural environments. The algorithm's robustness and generalization ability were enhanced through dataset expansion techniques to simulate various illumination conditions. Comparative results demonstrated that the average precision (AP) of sugarcane stem node detection using YOLOv4 was 95.17%, surpassing the performance of four other algorithms (with AP values of 78.87%, 88.98%, 90.88%, and 92.69%, respectively). Moreover, the detection speed of the YOLOv4 method reached 69 frames per second (f/s), exceeding the real-time detection requirement of 30 f/s. Huang et al. [20] employed a rectangular template moving horizontally across the sugarcane image with a specific step length. They calculated the average gray value on the G-B component image and determined the stem node position based on the maximum average gray value. However, the recognition rate of this method was affected by the step length and template width, resulting in a recognition rate of 90.77%. Xiao and Xu [28] constructed a sugarcane seed cutting recognition system based on deep separable convolution neural network. The system can identify sugarcane buds in the sugarcane planting and cutting process, so that they will not be damaged, thus reducing the rate of the injured buds and improving the cutting quality. In the work conducted by Lu et al. [29], they performed segmentation of the sugarcane image using the Herpes simplex virus (HSV) color space. This was achieved by applying a threshold to obtain a composite image, which was created by adding the inverse image of the H component and the S component image. Subsequently, a support vector machine (SVM) was employed to classify and recognize the blocks within the composite image. The average recognition rate of stem nodes achieved in this study was 93.36%.

There are still many problems that limit the use of machine vision, machine learning, deep learning, wavelet analysis, image processing algorithms and Herpes simplex virus (HSV) color space in cutting sugarcane seeds, as stated by [5, 21, 23, 30, 31]. These problems include slow speed, poor real-time performance, low identification efficiency, and high maintenance and operation costs. Sugarcane leaves must be removed to expose only the sugarcane stem, which consists mainly of the internode and stem node area, so that the machine can determine the location of the nodes, which represents an additional cost and can lead to the bud's damage if done incorrectly. In addition, the machine does not differentiate between a good bud from a damaged or injured one. Although scientists have made tremendous advances, there are still certain gaps in these studies.

To overcome the problems related to the application the other node detection systems in the process of cutting sugarcane seeds, the current study aims to design a new automatic sugarcane seed cutting machine based on internet of things (IoT) technology and RGB color sensors. The use of IoT and RGB color sensors achieved a high analytical performance without requiring the use of computers and high-definition high-speed camera for image processing like other automatic sugarcane seed cutting systems.

## 2. Materials and methods

### 2.1. System composition and working theory

The technology of preparing the sugarcane stem includes the sequential removal of leaves, preserving the internode zone, since this zone affects the growth and future development of buds

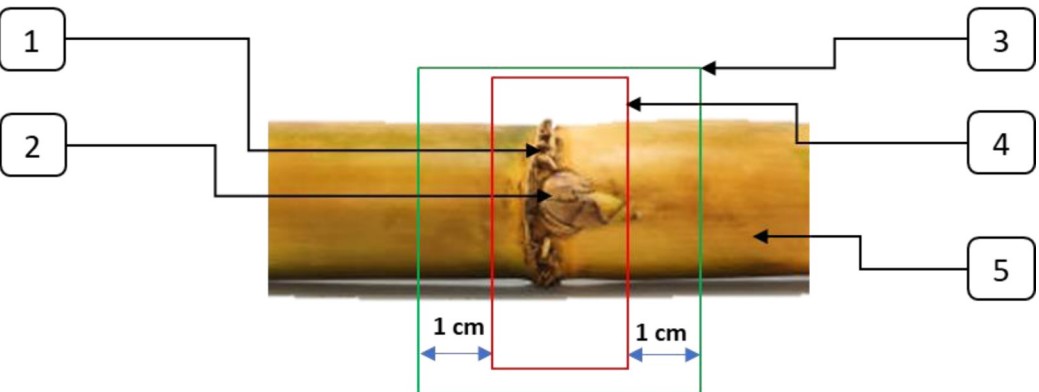

**Fig 1. Sugarcane-cutting agricultural requirements.** 1. leaf scar; 2. sugarcane buds; 3. sugarcane seed; 4. node area; 5. internodes area.

and seeds. This is due to the fact that the water and various nutrients necessary for the growth of each bud come from the internodes. Therefore, it is important to leave at least 1.0 cm of internodes both above and under the sugarcane bud when cutting seeds. Using this approach, there are enough resources left for the sugarcane buds to germinate [21, 26, 32]. The established requirements for cutting sugar cane are shown in Fig 1.

## 2.2. Composition of the developed machine

The prototype of the automatic sugarcane seed cutting machine (ASSCM) proposed in the current study was designed and manufactured in the Agricultural and Biosystems Engineering Department, Agriculture and Natural Sources Collage, Aswan University, Egypt. The main components of the ASSCM are shown in Figs 2 and 3. The ASSCM is associated with a special sugarcane feed system (conveyor belt), a sugarcane node scanning and detection system, node cutting system, sugarcane seed counting and monitoring system and a system that controls the subsequent management and final completion of the cutting process. The machine frame is made of steel with dimensions of 100 cm in length, 75 cm in height and 40 cm in width, as shown in Fig 4.

**2.2.1. Sugarcane feeding approach.** As the sugarcane stem feeding system is one of the most important processes of producing high-quality sugarcane seeds, a stepper motor was used to operate the transmission. The working theory of the feeding system is based on the passage of a sugarcane stalk between three groups of plastic rollers with the forces of friction generated between the sugarcane stalk and the rollers. Each group of rollers consists of an upper roller that is loaded on its coil springs and a lower pulley mounted on a ball bearing. The main components of the sugarcane feeding system and operation principle are shown in Figs 5 and 6.

**2.2.2. The sugarcane node detection system.** The sugarcane node detection system is mainly composed of two parts: a node scanning system, and a control system. The sugarcane node detection system consists of a pair of RGB color sensors (model: TCS3200 with Focusable lens) that measure the RGB color channels of the sugarcane stalks to determine the location of the hand-colored sugarcane nodes and send the scanning signals to the Arduino board (model: mega 2560) to process the data and then send an order to the cutting unit to cut the sugarcane stalks. Fig 7 shows 8 possible contacts of the RGB color sensor connected to the Arduino mega board.

Here, an RGB color sensor (model: TCS3200 with Focusable lens) is proposed as a candidate for the hand-colored sugarcane node detection. This and other similar sensors have been

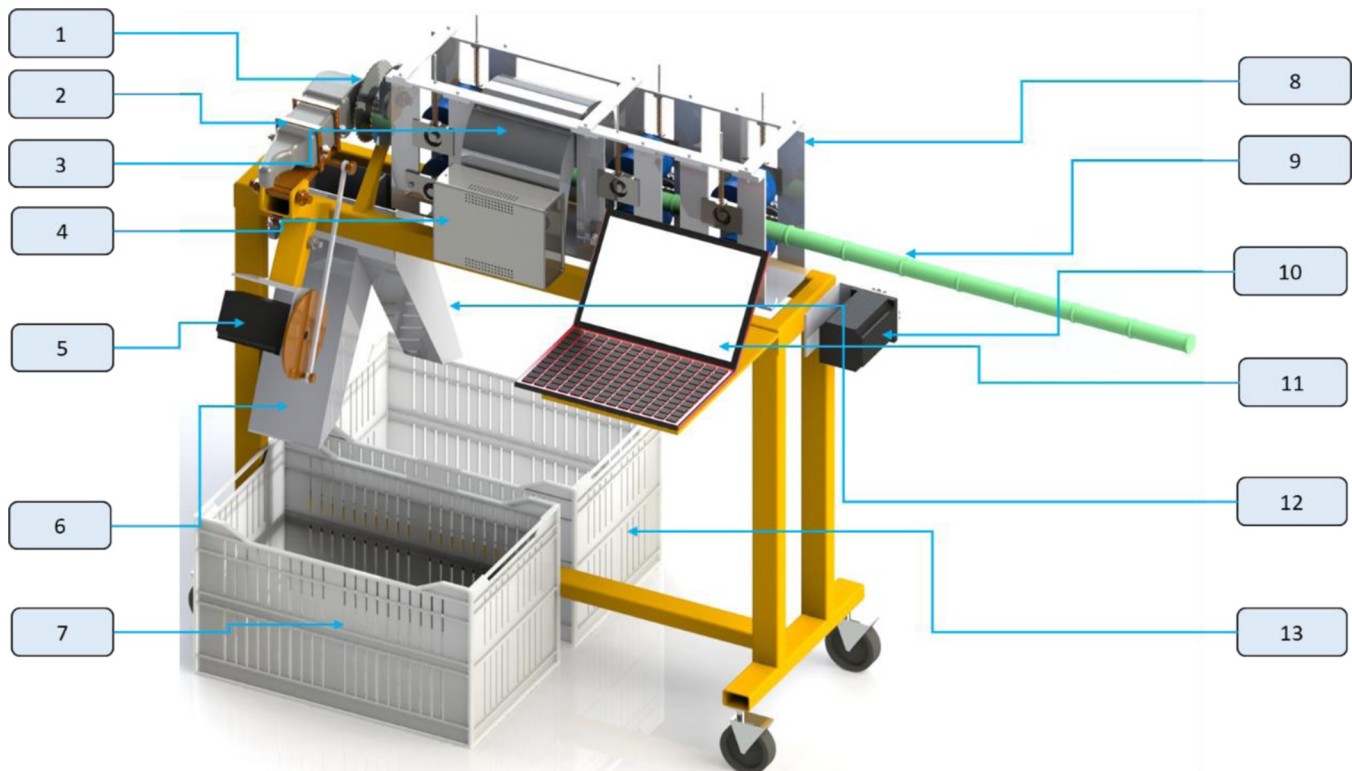

1. Saw knives; 2. Cutting motor; 3. Scanning zone; 4. Control system; 5. Stepper motor for cutting system; 6. Output pass (internodes); 7. Collecting Box (internodes); 8. Feeding system; 9. Sugarcane stalk; 10. Stepper motor for feeding system; 11. Laptop; 12. Output pass (nodes); 13. Collecting Box (nodes).

**Fig 2. 3D model showing the main components of the proposed machine.**

used by other authors to sort fruits [33], monitor plant leaf color as a plant status indicator [34], measure olive oil [35], wine [36], and banana [37] color for quality control purposes, or colorimetric gas detection [38]. The RGB color sensor proposed in this study, whilst inexpensive, was designed to accurately measure the color of a hand-colored sugarcane node and provide the red (R), green (G), and blue (B) coordinates in the RGB color space.

**2.2.3. The sugarcane node cutting system.** Fig 8 illustrates the sugarcane node cutting system, which enables the transmission of sugarcane stalks along the X-axis and the reciprocating movement of the cutting system along the Y-axis to carry out the cutting process of sugarcane nodes. The X-axis stepper motor (Nema 23) and its driver (TB6600) control the rotation of the plastic rollers through chains and sprockets, facilitating the movement of sugarcane stalks with hand-colored nodes in the positive X-axis direction for delivery and feeding. The cutting system includes a crank pulley mechanism, which is connected to another stepper motor (Nema 23) and its driver (TB6600). The reciprocating motion of the cutting system in the positive Y-axis direction, along with the high-speed rotation of symmetrical circular saw knives, achieves the cutting of the hand-colored nodes on the sugarcane stalks. By employing this configuration, the sugarcane node cutting system combines the synchronized movements of the X-axis transmission and the Y-axis cutting process, facilitated by the stepper motors and drivers mentioned above.

Sugarcane stems are composed of internodes and nodes, and within the sugarcane stalk, there are sugarcane buds and leaf marks, as depicted in Fig 1. To meet the requirements of the

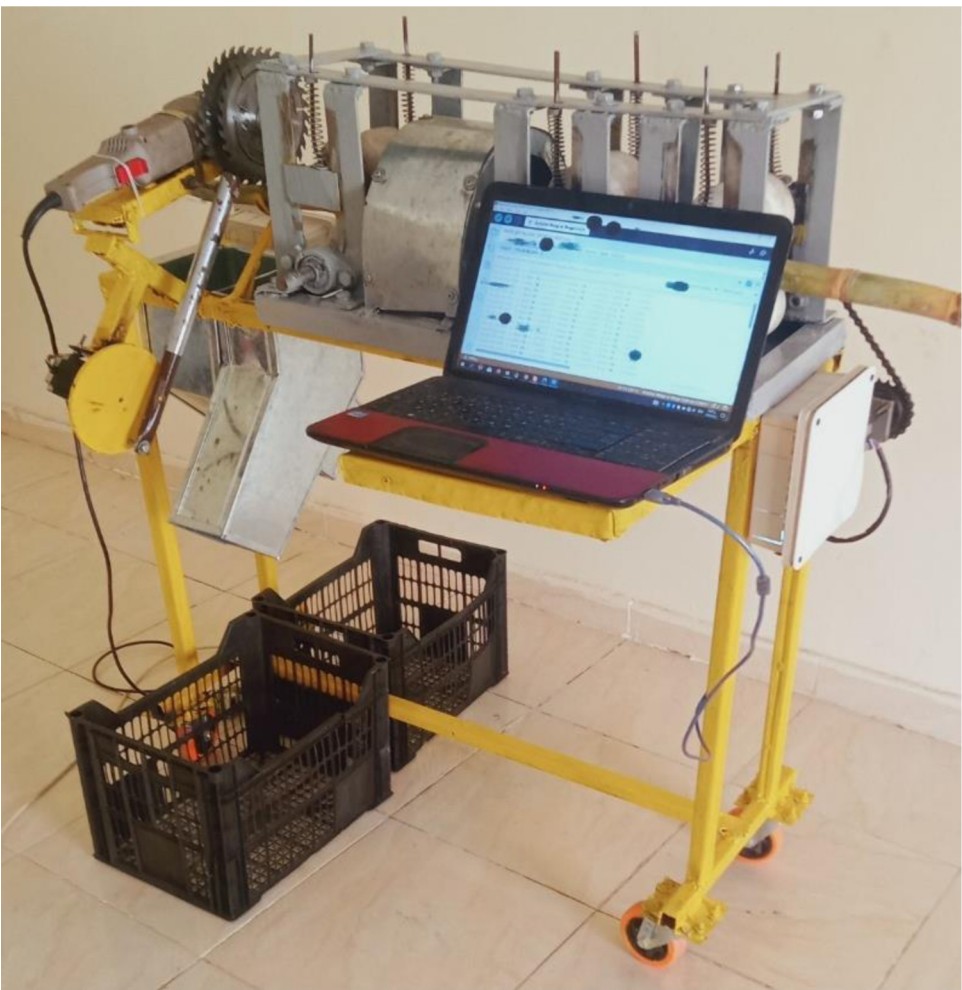

**Fig 3. Laboratory test of the ASSCM prototype.**

sugarcane seed cutting process, each individual sugarcane seed should have only one sugarcane node, with 1.0 cm long internodes on either side of the seed. These internodes provide the necessary nutrients for the subsequent growth of the sugarcane seed. To fulfill these specifications, Fig 8 showcases the design of a high-speed DC motor operating at 12,000 rpm, along with two symmetrical circular saw knives. To obtain the desired length of sugarcane seeds, a distance between two saw blades is left, approximately 3 cm. This distance can be adjusted.

The cutting forces between the sugarcane stalk and the circular saw knives were studied by Wang et al. [26], where it was stated that, when the unit is working, the circular saw blade rotates at high speed. To facilitate the study of the force on which sugarcane is cut, assuming that the cross-section of the sugarcane stalk is a regular circle, the direction returned by the disc knife is the positive direction of the $\xi$ axis, the vertically upward direction is the positive direction of the axis, ignoring the friction in the $\xi$ direction and the $O\xi\psi$ fixed coordinate system is established with the center $O$ of the disc knife as the origin. The force analysis diagram is shown in Fig 9.

The mechanical equilibrium Eqs (1 and 2) are listed according to the force applied to the sugarcane [26]:

$$F_{N1} + F_t \times \cos\theta - F_n \times sin\,\theta = 0 \tag{1}$$

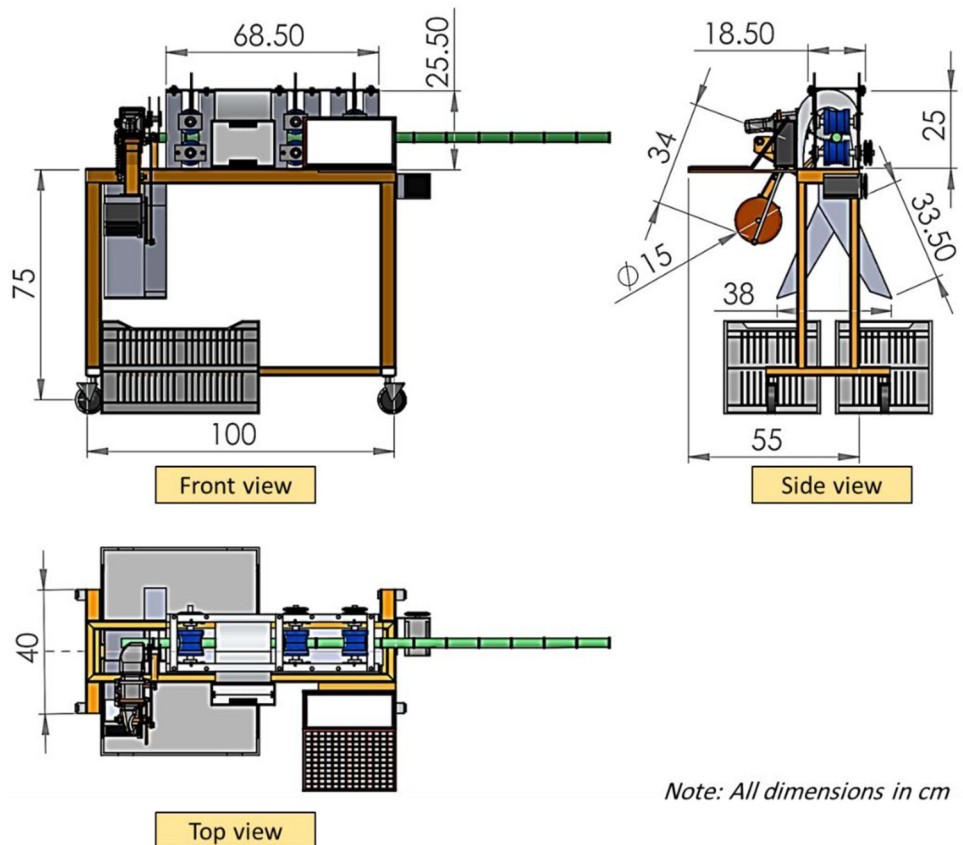

**Fig 4. Three main views of the ASSCM show the main dimensions.**

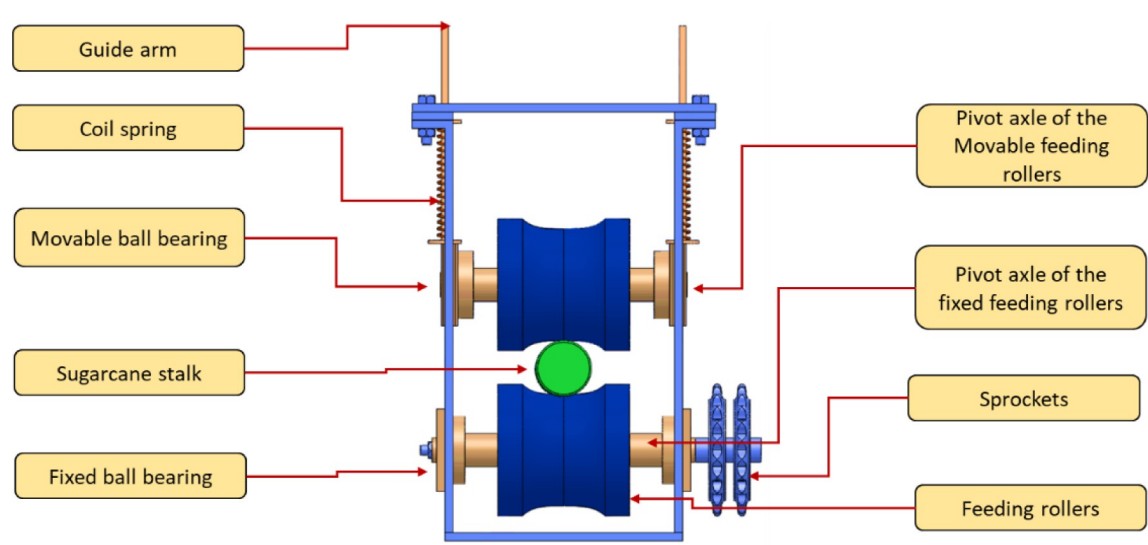

**Fig 5. Main components of the sugarcane feeding mechanism.**

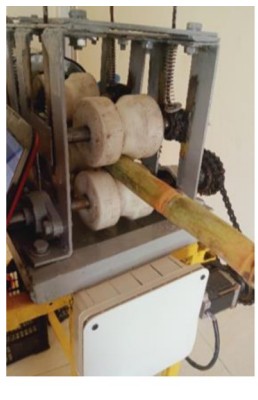
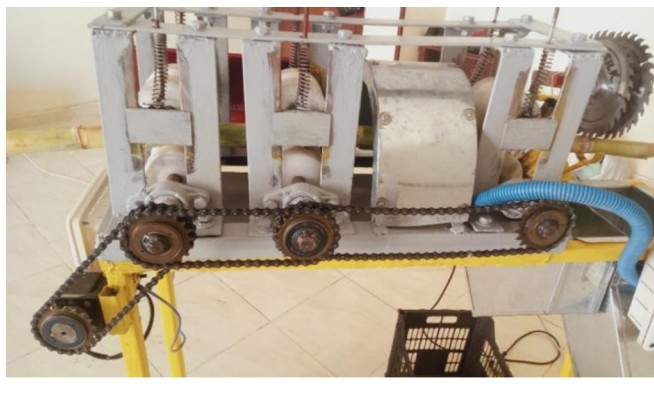

**(a)**                              **(b)**

**Fig 6. The sugarcane feeding system of the ASSCM.** (a): Sugarcane stalk between the upper and lower plastic rollers; (b) Side view of the feeding system showing the power transmission system from the stepper motor to the three lower plastic rollers.

$$F_{N2} - G - F_t \times cos\ \theta - F_n \times sin\ \theta = 0 \qquad (2)$$

where: $F_t$ is the tangential force of the circular saw knives on the cutting site, N; $F_n$ is the positive pressure exerted by the circular saw knives on the cutting site, N; $G$ is the sugarcane gravity, N; $F_{N1}$ is the supporting force of side baffle on sugarcane, N; $F_{N2}$ is the supporting force of the ground against the sugarcane, N; and $\theta$ is the angle at which the blade cuts the sugarcane.

**2.2.4. Sugarcane seed counting and monitoring unit (SSCMU).** The SSCMU consists of an ultrasonic sensor (model: HC-SR04) that is installed in the sugarcane seed exit path. When the sugarcane seeds pass through the exit path, they cut off the ultrasonic waves and thus this signal is sent to the control unit to process the data, and then the data is sent to the laptop via Wi-Fi module (model: Esp-8266), as shown in Fig 10.

The ultrasonic pulses travel outward until they encounter the sugarcane seeds. This option is shown in Fig 10. The sugarcane seeds cause the ultrasonic pulses to be reflected towards the receiver. The ultrasonic receiver would detect the reflected ultrasonic pulses and time consumed, with the speed of the ultrasonic waves 340 m/s in air. Based on the time consumed, the distance can be calculated between the sugarcane seeds and the transmitter. The time, rate, and distance (TRD) measurement formula are expressed in Eq 3 [40].

$$d = C \times T \qquad (3)$$

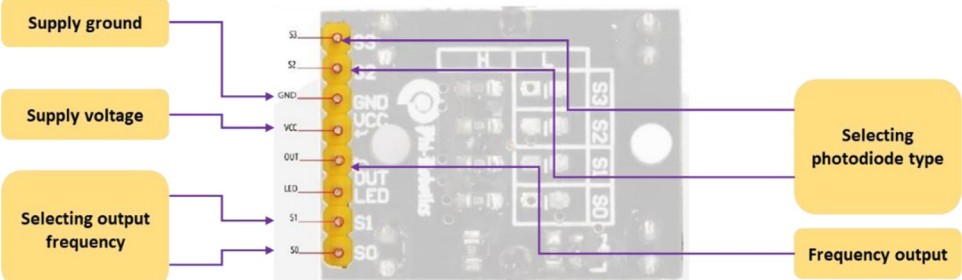

**Fig 7. Different pins of the color sensor [39].**

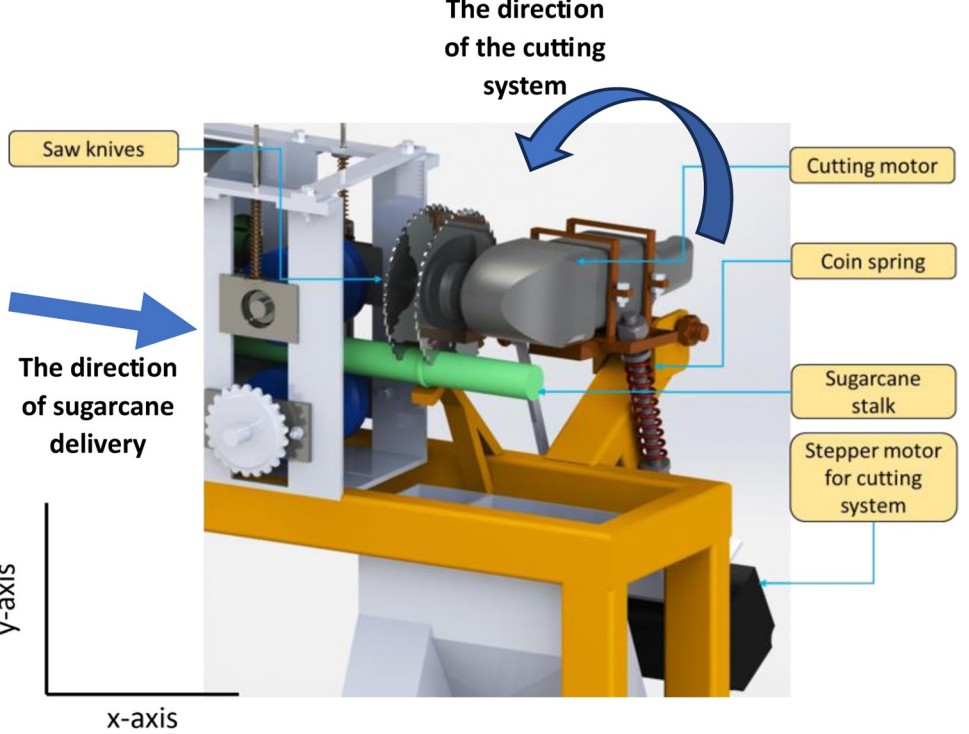

**Fig 8. Main components of sugarcane node cutting system.**

In this case, T is divided by 2, as T is double the time value from the transmitter to the sugarcane seeds and back to the receiver.

## 2.3. Operating principle

The operational procedure for cutting sugarcane seeds is displayed in Fig 11. Prior to uploading the programming code to the Arduino board, the code is modified to suit specific operational requirements. This includes adjusting parameters, such as the speed of the feeding system, the delay between scanning and cutting, the position of the color sensor lenses, and the speed of the cutting system. Once the modifications are made, the programming code is uploaded to the Arduino board. At the start of the operation, the stepper motors and color sensors are initialized. The sugarcane stalks are placed in the space of the feed system, then pushed inside where they are captured by the upper and lower rollers. The supply of sugarcane stalks inside is carried out due to the step-by-step movement of the engine along the  axis, controlled by the control system. As the sugarcane stalk with colored nodes passes through the scanning zone, the output signal from the color sensors is sent to the Arduino board for processing and decision-making based on pre-set values. There are two possible decisions based on the output signals. First, if the output signals match the pre-set value, the Arduino board issues a command to stop the feeding system and activate the cutting system. Second, if the output signals do not match the pre-set value, the feeding system continues to operate, while the cutting system remains in standby mode. The sugarcane seeds fall down the output pass and are collected in a designated box. Under normal conditions, when the output pass is empty, the ultrasonic sensor measures a distance of approximately 15 cm. However, when a sugarcane seed passes through the output pass, it interrupts the ultrasonic waves, resulting in a measured distance of less than 15 cm. The sugarcane seed counter sensor generates an electronic impulse upon

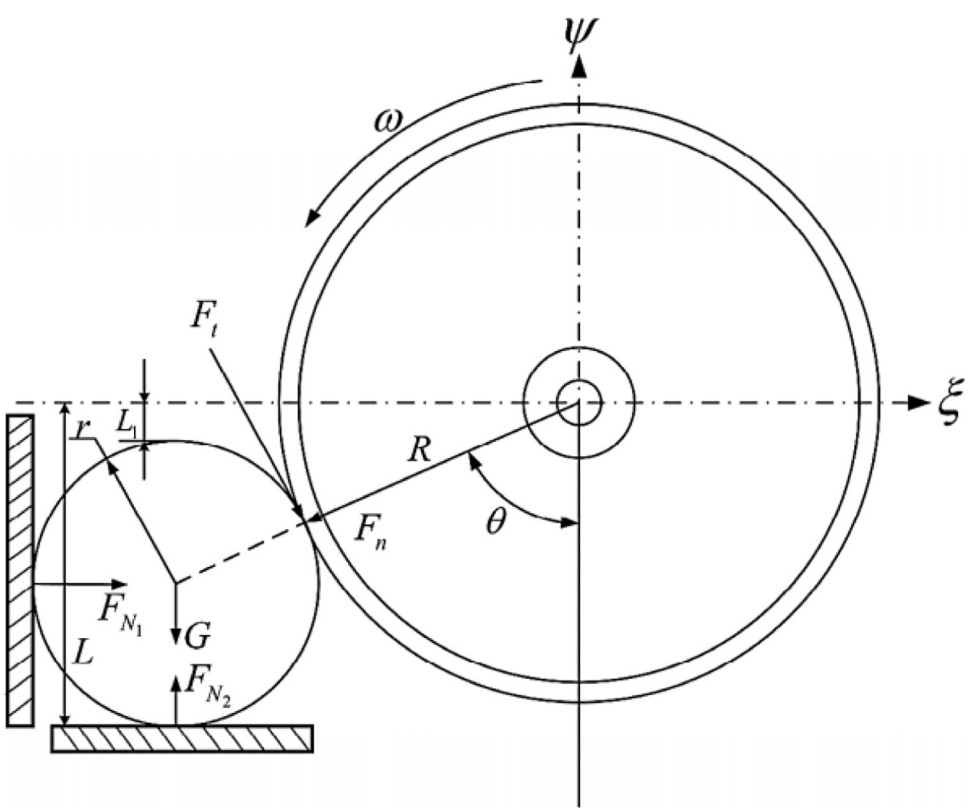

**Fig 9. Static stress analysis of sugarcane cutting [26].**

detecting the sugarcane seed, and the software program counts the seeds accordingly. The final count is then transmitted to the communication serial port, following the communication protocol. The communication serial port is connected to a Wi-Fi module (ESP-8266), which sends the collected data to a laptop software for real-time display.

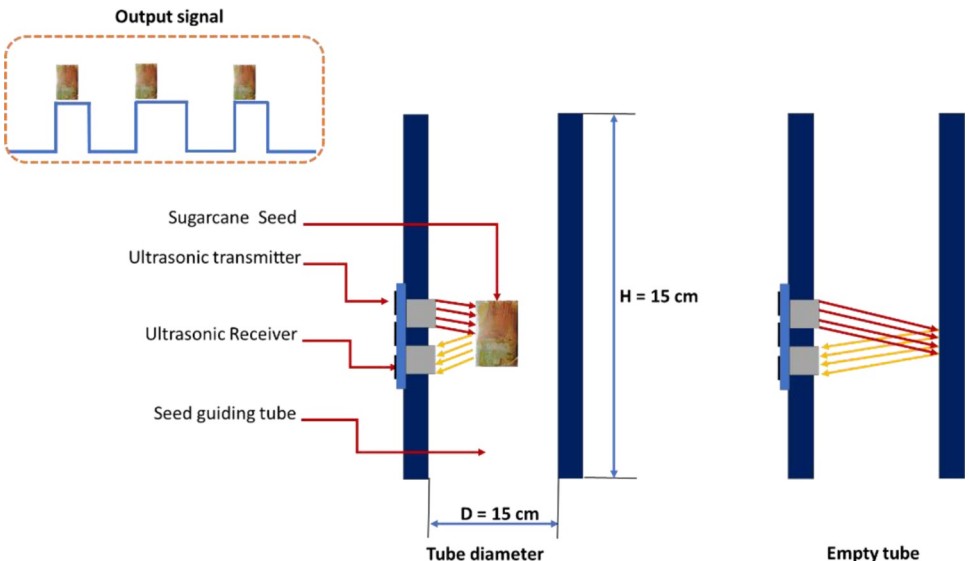

**Fig 10. Principle of operation of the SSCMU.** H is the height of the sugarcane seeds guiding tube, and D is the diameter of the guiding tube.

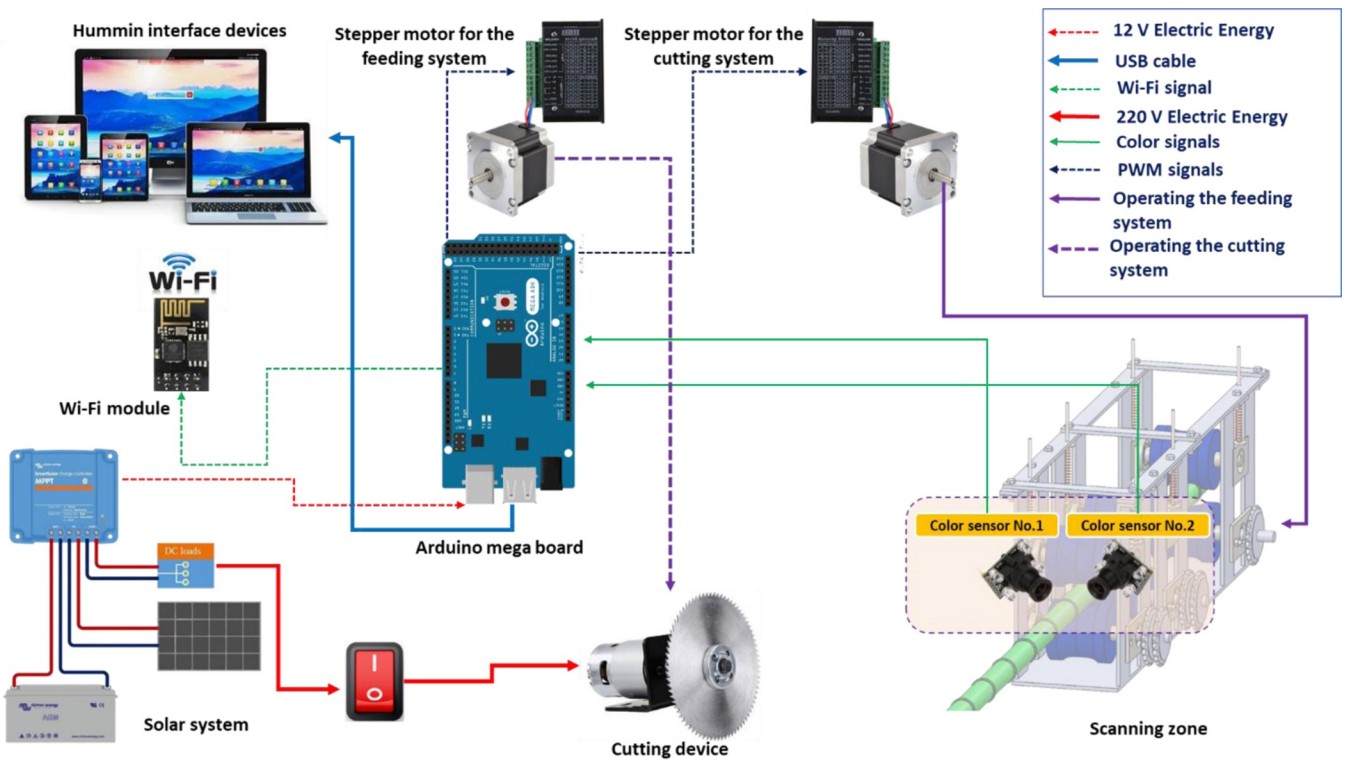

**Fig 11. The working process of cutting sugarcane seeds.**

## 2.4. Programming codes of RGB sensors

This section focuses on programming the operating codes used for detecting the RGB color characteristics. The process of programming the working codes used is presented below:

**Step 1:** It is necessary to determine the points of the RGB color sensors [S1, S2, S3 and S4] and combine them with three pulse width modulation variables in accordance with the desired color.

**Step 2:** The select contacts [S1, S2, S3 and S4] must be set as output contacts, since this process is responsible for increasing or decreasing the power of the color photodiode. The TCS3200 output is used as input.

**Step 3:** The RGB color sensor has contacts S2 and S3, they are used to determine the color characteristics. RGB color sensors are presented in the technical data sheet, their data is presented in Table 1.

Scaling the output frequency helps enhance the sensor analyses for different frequency counters or microcontrollers. The OUT pin provides frequency, which is selected as 20% initially. First of all, the data transfer rate is set at 9600 bytes per second, and after that, serial

**Table 1. Photodiodes characteristics [33].**

| S2 pin | S3 pin | Photodiode type | Output frequency scaling |
|--------|--------|-----------------|--------------------------|
| L | L | Red | Power down |
| L | H | Blue | 2% |
| H | L | Clear (no filter) | 20% |
| H | H | Green | 100% |

communication between the Arduino super board and the laptop is started using a USB cable. Writing to S2 [LOW] and S3 [LOW] activates the red photodiodes to take the readings for red color density and print the RGB color frequency. Writing to S2 [LOW] and S3 [HIGH] activates the blue photodiodes to take the readings for red color density and print the RGB color frequency. Writing to S2 [HIGH] and S3 [HIGH] activates to obtain red density results, blue photodiodes are used, and then the RGB color frequency is printed.

**Step 4:** When the node color detection is completed, the system starts sequentially and sets the RGB color value (the maximum or minimum RGB value of the output frequency is calculated in stages, and then stored with subsequent application).

## 2.5. Engineering factors affecting the performance of the RGB color sensors

The output PW are recorded taking into account different studied parameters (three color types for hand-colored nodes [black, red and blue]; three speeds of the feeding system [7.5 m/min, 5 m/ min and 4.3 m/ min]; three installing heights of the RGB color sensors [2.0 cm, 3.0 cm and 4.0 cm]; and three widths of colored line [10.0 mm, 7.0 mm and 3.0 mm].

Ma et al. [41] and Afrisal et al. [42] recommended setting the light intensity to 40 lux, as they found that higher light intensity increases reflectance and decreases RGB color pulse width (PW). Afrisal et al. [42] determined that fluctuations in lighting conditions do not significantly impact the system's accuracy, if light intensity is sufficient (between 150 and 500 lux). However, it is worth noting that machine vision systems can be influenced by both the level and quality of illumination, and changes in light levels and incorrect positioning may adversely affect the software used in the system, as mentioned by Brosnan et al. [43].

The 1.5-meter sugarcane stalk was put on the feeding system at zero time. Then, the feeding system was run at the feeding speed of 4.3 m/min, and the output pulses of the RGB color sensors changed up and down based on the color of the sugarcane stalk nodes and internodes. The output signals or pulses for the first four hand-colored nodes were recorded. Then, the output pulses for each test were collected and graphically presented to compare the effect of different engineering factors on the performance of RGB color sensors.

**2.5.1. The effect of color type on the efficiency of all RGB color sensors.** To identify the optimal color type used for coloring the sugarcane stalk nodes, the speed of the feed system was set at about 4.3–4.4 m/min, considering the specific intensity of lighting (in our case, it is 40 lux), and RGB color sensors were also installed. All tests were performed with each inhalation of three different colors (for example, black, red, and blue), as shown in Fig 12. Ten samples were tested 3 times. Medium sugarcane stalks with an average diameter of 2.72 cm were used on the tests and the sugarcane stalk nodes were hand-colored with a color line 10 mm in width. This test aims to determine the appropriate color type that gives the best RGB values to raise the node detection efficiency.

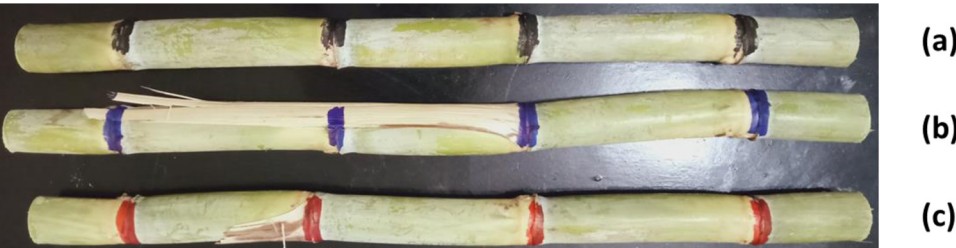

**Fig 12. Three sugarcane stalk samples with hand-colored node.** (a): Black hand-colored nodes; (b): Blue hand-colored nodes; and (c): Red hand-colored nodes.

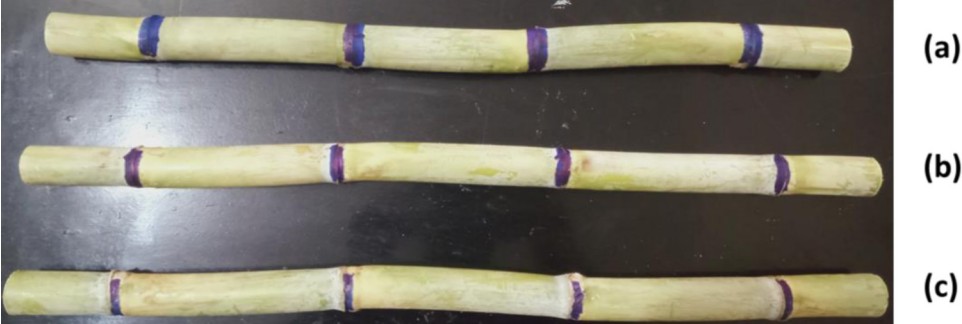

**Fig 13. Three sugarcane stalk samples with hand-colored nodes.** (a): Blue hand-colored nodes with 10 mm color width; (b): Blue hand-colored nodes with 7 mm color width; and (c): Blue hand-colored nodes with 3 mm color width.

**2.5.2. The effect of the feeding system speed on the performance of RGB color sensors.** The speed of the feeding system during the scanning and detection of hand-colored nodes to be cute can affect the quality and accuracy of the cutting process. For testing the effect of the feeding system speed on three different color channels (R channel, G channel, and B channel), three different speed of the feeding system of 7.5 m/min, 5.0 m/min and 4.3 m/min were tested. Ten sugarcane stalk samples were tested three times. The light intensity was adjusted at 40 lux, the RGB color sensors were fixed at 20 mm heigh. The nodes of sugarcane stalks with average diameter of 2.72 cm hand-colored with blue were used on the tests and the stalk nodes were colored with a color line 10 mm in width.

**2.5.3. The effect of the color sensors height on the performance of the RGB color sensors.** All experiments were conducted with three different heights of the RGB color sensors of 2.0 cm, 3.0 cm and 4.0 cm. Ten sugarcane stalk samples were tested three times. The speed of the feeding system was set at 4.3 m/min, light intensity at 40 lux, the RGB color sensors were fixed at 20 mm height. The sugarcane stalks with an average diameter of 2.72 cm were hand-colored with blue with a color line 10 mm in width and were used on the tests.

**2.5.4. Effect of width of colored line on the performance of the RGB color sensors.** The width of the color line has a vital role in the detection of the colored sugarcane stalk nodes. To identify the optimal width of the colored line used for coloring the sugarcane stalk nodes, three different widths of the colored line of 10 mm, 7.0 mm and 3.0 mm were used, as shown in Fig 13. The speed of the feeding system was configured, as shown in the example above, at a speed of 4.3–4.4 m/min, the lighting intensity was 40 lux, and the RGB color sensors were fixed at 20 mm height. The nodes of sugarcane stalks with an average diameter of 2.72 cm were hand-colored with blue to use on the tests.

## 3. Results and discussion

### 3.1. Effect of color type on the three-color channels (R, G and B) of the RGB color sensor

Black, red and blue hand-colored nodes are being scanned by two RGB color sensors. The graph tracks the sensor's response across three color channels: red (R), green (G) and blue (B). The y-axis represents the color pulse width (PW), indicating the strength or intensity of the color signal detected and the x-axis represents the scanning time based on the speed of the feeding system, measuring the duration of the sensor's readings.

**3.1.1. Black hand-colored nodes.** Fig 14 shows that lower PW values generally correspond to darker colors (like black), while higher values suggest brighter colors. The specific

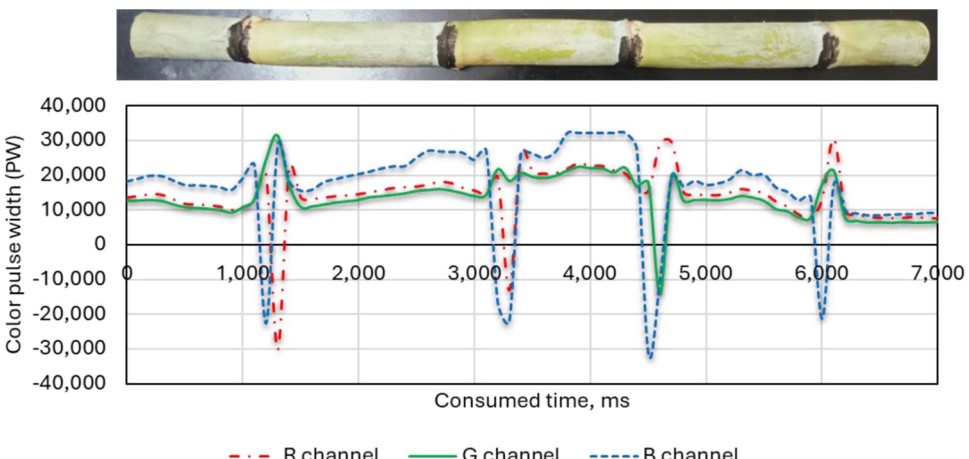

**Fig 14. Effect of black hand-colored nodes on the three-color channels (R, G and B) of the RGB color sensor.**

patterns and fluctuations in the graph reveal how the sensor responds to the black nodes across time and RGB color channels. Specific observations (based on typical RGB sensor behavior): 1. All three-color channels (R, G and B) are likely to exhibit relatively low PW values for black nodes, as black absorbs most light. 2. There might be slight variations in PW between channels due to sensor sensitivity and color filtering. 3. The shape of the curves could indicate the sensor's response time and accuracy in detecting black.

**3.1.2. Red hand-colored nodes.** As shown in Fig 15, all three-color channels (R, G and B) have high pulse width when the sensor is scanning red nodes. This is because red light is reflected strongly by the red node, so the sensor detects a strong signal in all three channels. The B channel has the highest pulse width of all three channels. The R and G channels also have significantly high pulse widths, although they are lower than the B channel. This means that the sensor detects some red and green light, even though the node is red. This could be due to several factors, such as: 1. The filters in the sensor may not be able to completely block out all non-red light, so some green and blue light may still leak through. 2. There is ambient light present, it may be reflected by the red node and picked up by the sensor in all three channels. 3. Some red light may be scattered within the sensor and some of this scattered light may

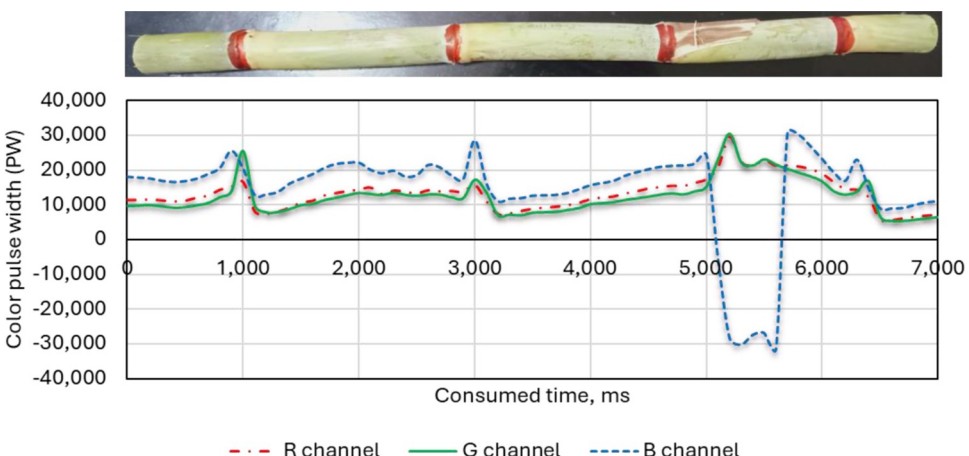

**Fig 15. Effect of red hand-colored nodes on the three-color channels (R, G and B) of the RGB color sensor.**

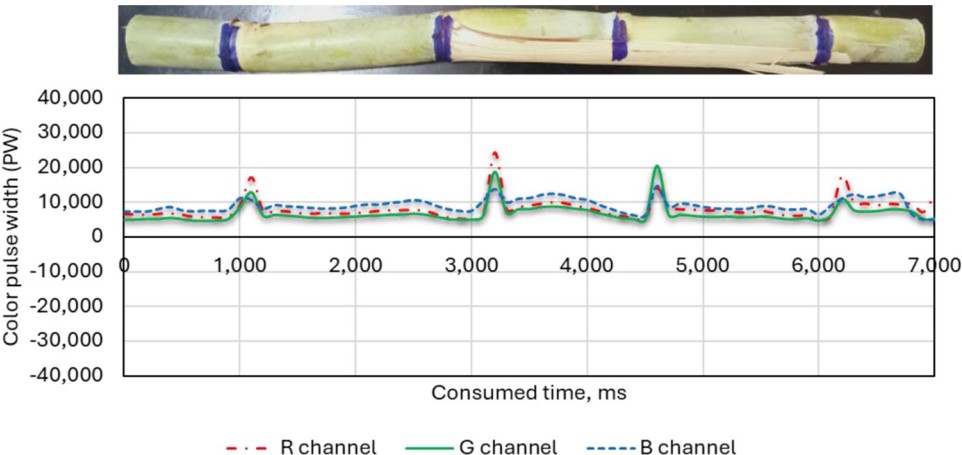

**Fig 16. Effect of BLUE colored nodes on the three-color channels (R, G and B) of the RGB color sensor.**

reach the green and blue detectors. The specific values and patterns of the pulse widths in the graph can tell us more about the performance of the RGB color sensor. For example, the width of the peaks and the presence of any noise in the signal can provide information about the sensor's resolution and sensitivity.

**3.1.3. Blue hand-colored nodes.** As expected, the blue channel (B) has the highest pulse width throughout the scan, indicating the strongest response to the blue nodes, as illustrated in Fig 16. The green channel (G) shows a moderately high pulse width, suggesting significant blue light leakage or reflection into the green channel. The red channel (R) has the lowest pulse width, close to the baseline, indicating minimal red-light detection from the blue nodes. The sensor effectively detects the blue color, as evident from the dominant blue channel response. However, there is significant interaction between the blue and green channels. This means some blue light is being detected by the green channel sensors as well. In addition, comparing the pulse width ratios between channels (B:G and B:R) for blue nodes to those for other colors (e.g., red nodes) would reveal the extent of interaction for different colors. Analyzing the noise levels in each channel could provide insights into sensor sensitivity and potential signal-to-noise ratio limitations. Overall, the graph demonstrates the RGB sensor's ability to identify blue nodes, but also highlights the presence of crosstalk between the blue and green channels. This information can be valuable for applications, where accurate color discrimination is crucial and might necessitate calibration or compensation techniques to address the crosstalk effect.

**3.1.4. Comparison between the three-color channels for the black, red and blue-colored sugarcane stalk nodes.** Fig 17 demonstrates the comparison between the three-color channels for the black, red and blue-colored sugarcane stalk nodes. The presented data show that black nodes had low PW values (minimal red light reflected), red nodes had high PW values (strong red-light reflection) and blue nodes had moderate PW values (some red-light leakage or crosstalk). The use of the red color to color the nodes of sugarcane stalks led to high fluctuation and instability in the width of the output pulses, which led to difficulty in monitoring and identifying the nodes of sugarcane stalks automatically. This fluctuation is since sugarcane stalks sometimes contain red because of insect infestations. It can also be that sugarcane stalks sometimes contain black color, although the pulse width resulting from using black color to color the nodes of sugarcane stalks showed good performance due to the used sticks being free of black color. However, when examining the pulse width resulting from the use of the blue

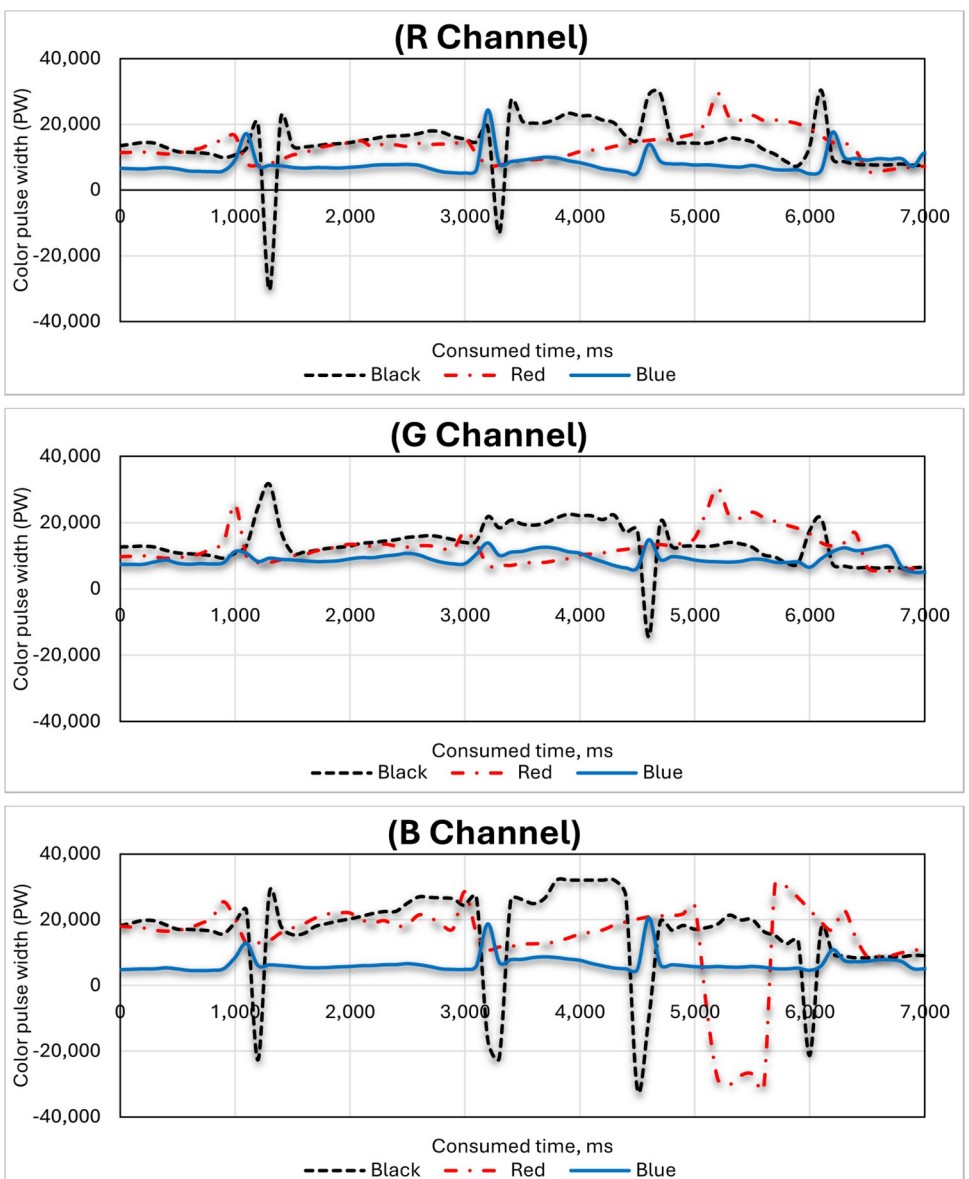

**Fig 17. Effect of colored nodes (black, red and blue) on the RGB color channels (R, G and B) of the color sensor.**

color, a perfect match is found that is free of any noise. Therefore, the blue color was used to color the nodes during the following tests. In the current study the yellow and green colors were deliberately excluded from the experiments because they represent the color of the dry and fresh leaves.

## 3.2. Effect of the feeding system speed on the three-color channels (R, G and B) of the RGB color sensor

Fig 18 shows the effect of the feeding system speed on the three-color channels (R, G and B) of the RGB color sensor, where three different speeds of the feeding system were used: $v1 = 4.3$ m/min, $v2 = 5.0$ m/min and $v3 = 7.5$ m/min. When comparing the pulse width coming out of different color channels of the RGB color sensors, it was found that an increased speed of the

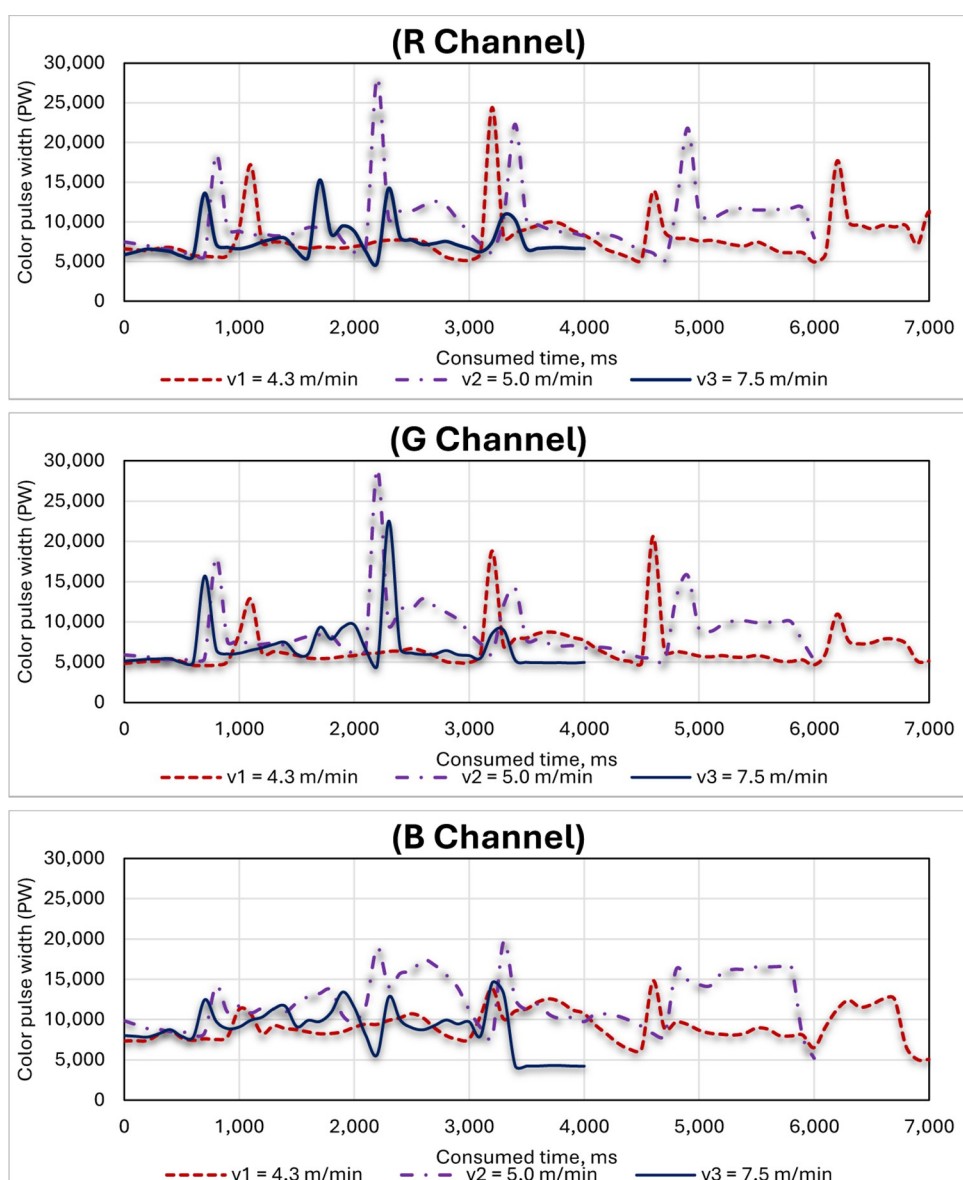

**Fig 18. Effect of the feeding system speed on the RGB color channels (R, G and B) of the color sensor.**

feeding system might lead to decreased pulse width across all channels due to shorter node scanning times, potentially affecting color detection accuracy. In addition, a faster feeding system might impact channels differently, potentially altering color balance or signal quality. When comparing the pulse width resulting from the three-color channels at different speeds of the feeding system, the red channel (R) was more stable than the rest of the color channels (G and B), followed by the green channel (G), while the blue channel (B) was the most fluctuating in the width of the pulses resulting from the scanning process of the blue nodes of the sugarcane stalks. Thus, programming the RGB color sensor involves relying on the red and green color channels only and avoiding the blue color channel. Huynh et al. [44] recommended that the speed of 7.2 m/min is considered as the maximum operating speed of the conveyor belt which can ensure the operation of the sensor.

### 3.3. Effect of height of the RGB color sensors on the three-color channels (R, G and B)

Fig 19 shows the effect of the height of the RGB color sensors on the three-color channels (R, G and B) of the RGB color sensors. The tests were repeated in each batch with three different heights of the RGB color sensors: 2.0 cm, 3.0 cm, and 4.0 cm. The obtained results showed that, as sensor height increases, pulse width generally decreases. This is primarily due to many reasons such as: 1. Light intensity decreases with the square of the distance from the source. Imagine a cone of light spreading from the sensor. At close range, the cone is concentrated, illuminating a smaller area with more intense light. As height increases, the cone widens, spreading light over a larger area and reducing intensity at any given point. 2. Increased height introduces more ambient light, potentially diluting the sensor's signal and reducing pulse

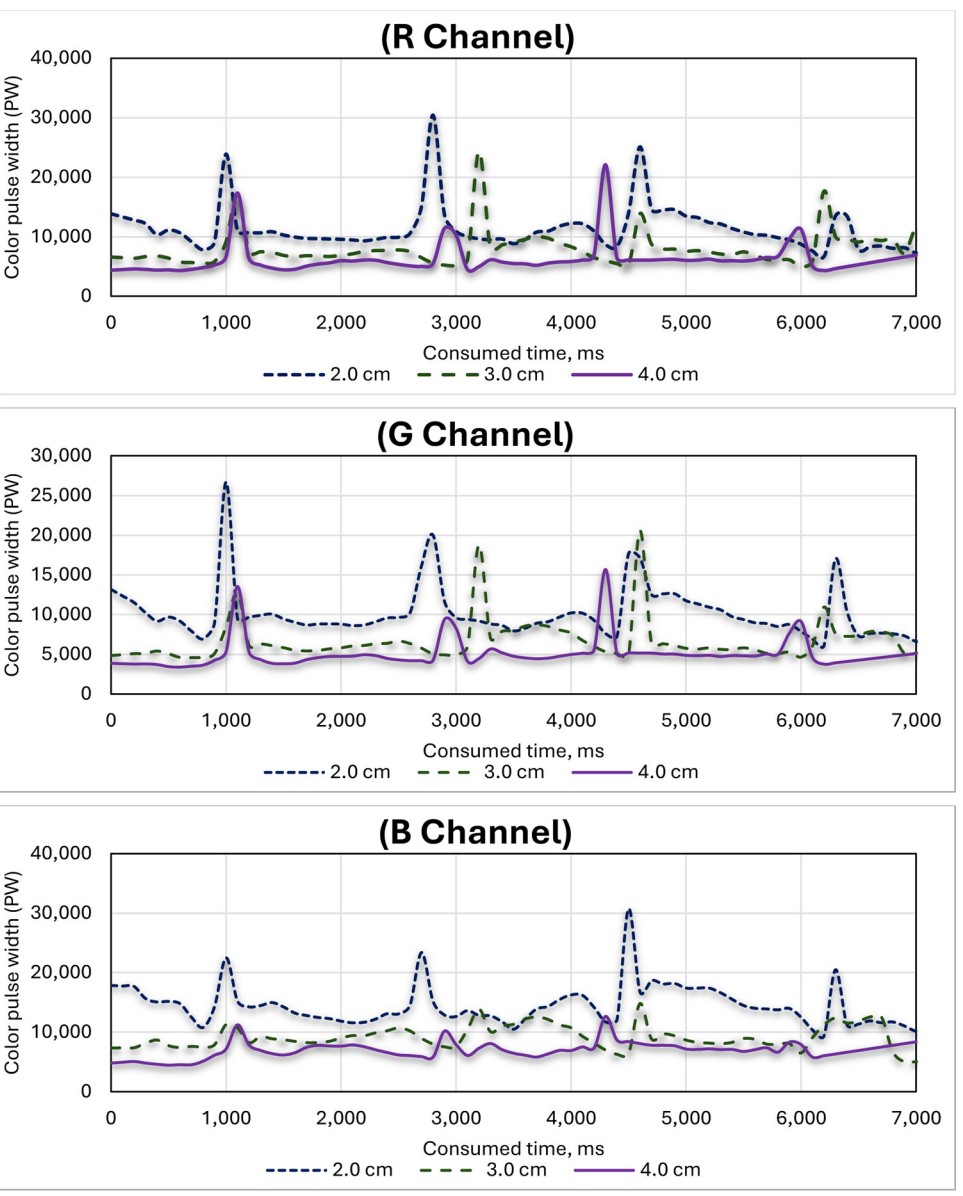

**Fig 19. Effect of height of the RGB color sensors on the RGB color channels (R, G and B) of the color sensor.**

width. 3. Specific sensor design and sensitivity also influence the effect of height. Therefore, in this study the best pulse width through the three-color channels of the color sensor was when the color sensors were installed at a vertical distance of 2.0 cm from the sugarcane stalk, followed by 3.0 cm, while the lowest pulse width was when the color sensors were installed at a vertical distance of 4.0 cm. According to the findings of Elwakeel et al. [33], it is recommended to set the height of the color sensor within the range of 15 to 30 mm. This adjustment enhances the sensor's capability to differentiate and discern the output signals effectively. Furthermore, Huynh et al. [44] noted that placing the sensor at a distance closer than 15 mm is not feasible due to spatial constraints.

### 3.4. Effect of width of color marker on the three-color channels (R, G and B) of the RGB color sensor

Fig 20 shows the effect of the width of the color marker on the three-color channels (R, G and B) of the RGB color sensors. Tests were run in every batch of three different widths of the color marker: 10 mm, 7.0 mm, and 3.0 mm. The maximum significant differences obtained for the width of the output pulse were from 10 mm color width, followed by 7 mm color width, while the lowest pulse width was obtained from 3 mm color width. This is because increasing the width of the coloring line increases the colored area that needs to be discovered and identified using the color sensor, thus increasing the possibility of identifying it.

### 3.5. Comparative analysis

Table 2 presents the comparison of the experimental results of the current study with those of other researchers around the world. laboratory experiments were performed to verify the reliability of the experiments and the correctness of the theory. The proposed system has a significant advantage over the other systems presented in literature because it can identify the sugarcane stalk nodes more accurately to help farmers produce sugarcane seeds at a very low cost compared to the other methods, with a short identification time and a recognition rate of up to 100%. Cooperation with factories will help to further optimize the system and mass production to contribute to agricultural mechanization and production.

## 4. Conclusions and future works

The final goal of the current study was to develop a new automatic sugarcane seed cutting machine (ASSCM) based on internet of things (IoT) technology and RGB color sensors. The use of IoT and RGB color sensors achieved a high analytical performance without requiring the use of computers or high-definition high-speed camera for image processing like other automatic sugarcane seed cutting systems. The developed ASSCM is coordinated by a sugarcane feeding system (conveyor belt), a sugarcane node scanning and exposure system, node cutting system, sugarcane seed counting and control and inspection system to complete the cutting operation. There are some variables that can affect the performance of the developed ASSCM. This research aims to study pulse duration (PW) of three-color channels (R, G and B) using RGB color sensors, where the output PW of red, green and blue channel values were recorded at different engineering factors (three color types for hand-colored nodes [black, red and blue]; three speeds of the feeding system [7.5 m/min, 5 m/ min and 4.3 m/ min]; three installing heights of the RGB color sensors [2.0 cm, 3.0 cm and 4.0 cm]; and three width of colored line [10.0 mm, 7.0 mm and 3.0 mm]). The obtained results from the laboratory tests showed that: 1. A perfect match free of any noise was found on the output pulse width of the sugarcane stalk nodes hand-colored blue compared with the other nodes colored black and

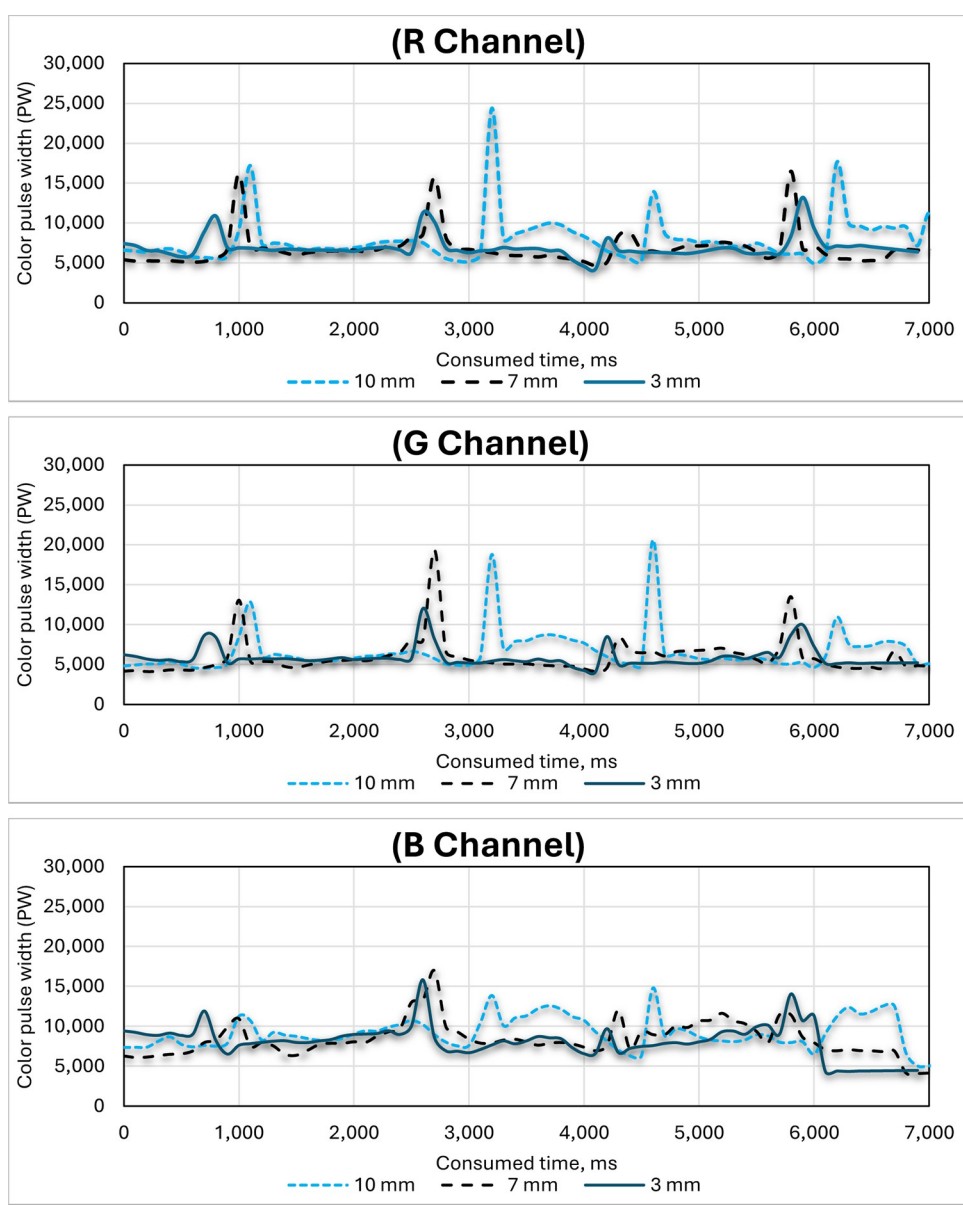

**Fig 20. Effect of width of color marker (10, 7 and 3 mm) on the RGB color channels (R, G and B) of the color sensor.**

red. Therefore, the blue color was used to color the sugarcane stalk nodes during the laboratory tests. 2. Increasing the speed of the feeding system led to decreased pulse width across RGB color channels due to shorter node exposure (scanning) times, potentially affecting color detection accuracy. 3. As sensor height increases, PW generally decreases. The highest PW was recorded at 2.0 cm sensor height, followed by 3.0 cm sensor height, while the lowest PW was recorded at 4.0 cm sensor height; 4. The maximum PW was obtained when coloring the sugarcane nodes with a blue color with a 10 mm color line, followed by 7 mm, while the minimum pulse width was obtained when coloring the sugarcane nodes with a blue color with a 3 mm color line.

**Table 2. Comparison of the obtained results of the RGB color sensors with other technologies.**

| Ref. | Technology | Detection time/node (sec) | Recognition rate, % | Owning and operating cost | The shape of cane sticks | Node Recognition |
|------|-----------|---------------------------|---------------------|---------------------------|--------------------------|------------------|
| [16] | Image processing and machine vision technology | 0.5 | 80 | High | Clean without leaves | Good and infected |
| [19] | Wavelet analysis | 0.21 | 98.5 | High | Clean without leaves | Good and infected |
| [21] | Machine vision | 0.539 | 93 | High | Clean without leaves | Good and infected |
| [23] | Wavelet analysis | 0.25 | 99.63 | High | Clean without leaves | Good and infected |
| [24] | Maximum value points of the vertical projection function | – | 95 | High | Clean without leaves | Good and infected |
| [27] | Wavelet analysis | 0.25 | 99.63 | High | Clean without leaves | Good and infected |
| [45] | Yolov$_3$ network | 0.028 | 90.38 | High | Clean without leaves | Good and infected |
| Proposed | Internet of thing (IoT) and RGB color sensors | 1.0–1.75 | 95–100 | Low | With leaves | Good only |

## 4.1. Recommendations

To achieve the highest performance of the scanning and detection system, it is recommended to hand-color sugarcane stalks with a 10 mm blue color line and install the RGB color sensor at 2.0 cm in height, as well as adjust the speed of the feeding system to 7.5 m/min to obtain the highest detection efficiency of the scanning system (100%).

## 4.2. Future work

The outcomes of the present study serve as an initial step towards exploring the potential applications of a newly developed technology in sugarcane harvesting machines. Furthermore, it paves the way for the automation of such systems through the utilization of robotics. However, further investigations are essential to scrutinize the machine's characteristics, including the type of knives, cutting speeds, damage index, damage frequency, cutting efficiency, and economic viability. Such investigations will enable researchers to determine the machine's efficacy and identify the optimal parameters to enhance its performance and practicality in the industry.

## Author Contributions

**Conceptualization:** Daniel Eutyche Mbadjoun Wapet.

**Funding acquisition:** Daniel Eutyche Mbadjoun Wapet.

**Investigation:** Mohamed E. Badawy, Daniel Eutyche Mbadjoun Wapet, Manar A. Ourapi, Mahmoud M. Hussein.

**Methodology:** Daniel Eutyche Mbadjoun Wapet, Abdallah E. Elwakeel.

**Resources:** Liu Yang, Mohamed E. Badawy.

**Software:** Daniel Eutyche Mbadjoun Wapet, Abdallah E. Elwakeel.

**Supervision:** Loai S. Nasrat, Tamer M. El-Messery.

**Validation:** Tamer M. El-Messery, Irina Aleksandrova, Mohamed Metwally Mahmoud.

**Writing – original draft:** Liu Yang, Daniel Eutyche Mbadjoun Wapet, Manar A. Ourapi, Abdallah E. Elwakeel.

**Writing – review & editing:** Daniel Eutyche Mbadjoun Wapet, Mohamed Metwally Mahmoud, Mahmoud M. Hussein.

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
