## [Decision Letter · Decision Letter 0]

27 Feb 2024

PONE-D-24-05665A New Automatic Sugarcane Seed Cutting Machine Based on Internet of Things TechnologyPLOS ONE

Dear Dr. Mbadjoun Wapet,

Thank you for submitting your manuscript to PLOS ONE. After careful consideration, we feel that it has merit but does not fully meet PLOS ONE’s publication criteria as it currently stands. Therefore, we invite you to submit a revised version of the manuscript that addresses the points raised during the review process.

We look forward to receiving your revised manuscript.

Kind regards,

Sathishkumar Veerappampalayam Easwaramoorthy

Academic Editor

PLOS ONE

Journal Requirements:

2. We note that your Data Availability Statement is currently as follows: "All relevant data are within the manuscript."

Reviewers' comments:

Reviewer's Responses to Questions

**Comments to the Author**

1. Is the manuscript technically sound, and do the data support the conclusions?

Reviewer #1: Yes

Reviewer #2: Partly

Reviewer #3: Yes

2. Has the statistical analysis been performed appropriately and rigorously? 

Reviewer #1: No

Reviewer #2: No

Reviewer #3: N/A

3. Have the authors made all data underlying the findings in their manuscript fully available?

Reviewer #1: No

Reviewer #2: Yes

Reviewer #3: Yes

4. Is the manuscript presented in an intelligible fashion and written in standard English?

Reviewer #1: Yes

Reviewer #2: Yes

Reviewer #3: Yes

5. Review Comments to the Author

Reviewer #1: The manuscript entitled “A New Automatic Sugarcane Seed Cutting Machine Based on Internet of Things Technology” includes the design of a new automatic sugarcane seed cutting machine consisting of a feeding system, a node scanning and detection system, a node cutting system, a sugarcane seed counting and monitoring system, and a control system.

The design of the sugarcane seed-cutting machine is an important and interesting contribution to increasing the effectiveness and performance of the planting of sugarcane. Therefore, it is valuable scientific research that deserves to be published in a reputable international scientific journal.

MS is enjoyable, informative, and easy to understand. It contains potentially useful information for readers. All sections of MS are well organized and presented.

Following the general assessments mentioned earlier, you will find additional minor improvements listed below:

The abstract should begin with a short sentence emphasizing the importance of the topic.

The first paragraph of the Materials and Methods appears to be more suitable for the Introduction section. Please consider revising the placement for better coherence. In the second paragraph, include relevant citations to support the details provided. This will enhance the credibility of the research.

Address the lack of information on the statistical foundation of the paper. Clearly outline the statistical methods employed in the study to strengthen the research.

I couldnt find the relation of the study with Internet of Things Technology. Therefore title should be revised.

P.S. Writing a review report for a manuscript lacking line numbers poses challenges, making it difficult to provide specific feedback. Therefore, I recommend authors include line numbers before submitting to any journal.

Reviewer #2: The pulse widths (PW) of three color channels (Red-R, Green-G, and Blue-B) of the sensors under the laboratory conditions have been proposed in this study. The IoT and RGB color sensors are possible to get the analytical indicators similar to those achieved with other automatic systems for cutting sugar cane seeds without requiring using computers. This study is interesting however there are some drawbacks that the authors should address them to improve this study.

1.In the introduction, the contributions of this study must be provided clearly especially the proposed methods comparing with the existing studies.

2.At item 2.3, the authors stated that “The operational procedure for cutting sugarcane seeds is displayed in Figure 4.” This statement is inaccurate to describe this content.

3.In table 2, the authors compared with other technologies, based on the certain values, this technology is not benefit than previous technologies.

4.The scientific basis of combined methods between the internet of things (IoTs) and RGB sensors must be provided.

5.Its significance of this study is inadequate in the novelty methods.

6.The limitation of this study is not provided as well as its application in the food manufacturing

Reviewer #3: Below are my comments that may help the authors further improve their manuscript titled "A New Automatic Sugarcane Seed Cutting Machine Based on Internet of Things Technology":

1. Numbering the manuscript's lines would be better to facilitate the review process. Please consider this comment in the revised version.

2. Please revise the affiliation numbers for all authors.

3. P.3 L.17: Please replace Xiao et al. [29] with Xiao and Xu [29] because there are only two authors.

4. Please revise the numbers of figures (i.e., Figure 1 is iterated). Please address this comment and ensure the figures numbers are consistent with the manuscript's text.

5. Figure 3: Please revise and correct all dimensions in mm! I think you mean that all dimensions in cm.

6. The authors should follow the journal's guidelines in writing the headings (i.e., some headings are written capitalize each word and bold, and others do not). Please consider this comment throughout the manuscript's text.

7. Figure 7 is iterated; please revise it and ensure the figure numbers are cited correctly within the manuscript's text.

8. The authors should cite the equations' numbers within the manuscript's text. Please address this comment for all equations.

9. P.12 L.14: Please be uniform in writing the unit of lux throughout the manuscript's text.

10. P.12 L.16: Please revise and replace the literature review Brosnan et al. [41] with Brosnan et al. [42] as cited in the references list.

11. P12 L.32: Please revise the word dimeter throughout the manuscript's text.

12. P.13: Please revise the title of heading 3.2.2.

13. P.19, 20 L.3, 11: Please revise the literature review's authors with the one cited in the references list.

14. P.20 L.6: Please revise the value of the vertical distance of 1.0 cm.

15. What about the limitations of this study? The authors should mention the study's limitations within the manuscript's text. Please consider this comment.

6. PLOS authors have the option to publish the peer review history of their article (what does this mean?). If published, this will include your full peer review and any attached files.

Reviewer #1: **Yes: **Davut Karayel

Reviewer #2: No

Reviewer #3: **Yes: **Mahmoud Okasha

---

## [Author Response · Author response to Decision Letter 0]

6 Mar 2024

***Technical response to the reviewers*** March, 2024

Journal name: PLOS ONE 

Title: “A New Automatic Sugarcane Seed Cutting Machine Based on Internet of Things Technology and RGB color sensor”

Liu Yang 1, Loai S. Nasrat 2, Mohamed E. Badawy 3, Daniel Eutyche Mbadjoun Wapet 4,*, Manar A. Ourapi 5, Tamer M. El-Messery 6, Irina Aleksandrova 6, Mohamed Metwally Mahmoud 7, Mahmoud M. Hussein8 and Abdallah E. Elwakeel 5

1 School of Mechanical and Electrical Engineering, Shihezi University, Xinjiang Shihezi 832003, China, lyhake@163.com

2 Electrical Power Engineering Department, Faculty of Engineering, Aswan University, Aswan 81528, Egypt, loaisaad@yahoo.com

3 Agricultural Engineering Research Institute - Dokki – Giza 12611, Egypt, Mohamedelshahat@gmail.com

4&* National Advanced School of Engineering, Universit´e de Yaound´e I, Yaound´e, Cameroon, eutychedan@gmail.com

5 Agricultural Engineering Department, Faculty of Agriculture and Natural Resources, Aswan University, Aswan 81528, Egypt; Abdallah_elshawadfy@agr.aswu.edu.eg, Manarourpi@gmail.com

6 International Research Centre “Biotechnologies of the Third Millennium”, Faculty of Biotechnologies (BioTech), ITMO University, St. Petersburg, 191002, Russia, telmesseri@itmo.ru; ivaleksandrova@itmo.ru

7 Electrical Engineering Department, Faculty of Energy Engineering, Aswan University, Aswan 81528, Egypt, Metwally_M@aswu.edu.eg

8 Department of Communications Technology Engineering, Technical College, Imam Ja’afar Al-Sadiq University, Baghdad, 10053, Iraq, mahmoud_hussein@aswu.edu.eg

*Corresponding author: Daniel Eutyche Mbadjoun Wapet, eutychedan@gmail.com

Dear Editors and Reviewers

The authors are thankful to the learned Editors and Reviewers for their thoughtful and detailed comments to improve the quality of the manuscript. The authors have tried to address all the concerns, and the corrections are incorporated in the revised manuscript. The replies to the reviewer’s comments are provided below.

We hope that this revised version can meet the reviewer’s expectations and the standards for publication in the agriculture machinery Research.

The changes incorporated in the revised manuscript are highlighted in Yellow.

Editor's Comments:

Our sincere thanks and appreciation to the Editors for recommending the submission of the revised manuscript with major revision. To improve the quality of the manuscript, the reviewer's queries are addressed, and their suggestions are incorporated into the revised manuscript. 

Reviewer Comments:

Reviewer 1:

Comment-1: The abstract should begin with a short sentence emphasizing the importance of the topic.

Response-1: The authors are thankful to the honorable reviewer for the words of encouragement and trust in our work. We completely agree with you, kindly check the updated paper (Abstract section).

Comment-2: The first paragraph of the Materials and Methods appears to be more suitable for the Introduction section. Please consider revising the placement for better coherence. In the second paragraph, include relevant citations to support the details provided. This will enhance the credibility of the research.

Response-2: The authors are extremely thankful to the reviewer for this thoughtful point. We agree with you, kindly check the updated paper (Materials and Methods section).

Comment-3: Address the lack of information on the statistical foundation of the paper. Clearly outline the statistical methods employed in the study to strengthen the research.

Response-3: The authors are extremely thankful to the reviewer for this thoughtful point. But the laboratory experiments were undertaken with the specific objective of exploring the impact of several variables, such as the diameter of the cane sticks, the width of the coloring line, and the height of the RGB color sensors, on the width of the output pulse. These experiments did not warrant a statistical analysis. It's worth noting that several practical studies have been conducted on sugarcane cutting machine automation, but none have included statistical analysis. Such as, reference no. 5, 18, 20, 22, 23, 25

Comment-4: I couldn't find the relation of the study with Internet of Things Technology. Therefore, title should be revised.

Response-4: The authors wish to express their deep gratitude to the esteemed reviewer for their insightful comment. Following the reviewer's advice, we have revised the title of the manuscript and provided a more thorough explanation of the study's relationship to IoT. Specifically, we have added clarifying remarks and information in lines (242-245) and (278-282), as well as in Figure 11. Kindly check the updated paper.

Comment-5: P.S. Writing a review report for a manuscript lacking line numbers poses challenges, making it difficult to provide specific feedback. Therefore, I recommend authors include line numbers before submitting to any journal.

Response-5: The authors are extremely thankful to the reviewer for this thoughtful point. We completely agree with you. Kindly check the updated paper.

Reviewer 2:

Comment-1: In the introduction, the contributions of this study must be provided clearly especially the proposed methods comparing with the existing studies.

Response-1: The authors are extremely thankful to the reviewer for this thoughtful point. We agree with you, kindly check the updated paper.

Comment-2: At item 2.3, the authors stated that “ The operational procedure for cutting sugarcane seeds is displayed in Figure 4.” This statement is inaccurate to describe this content.

Response-2: The authors are extremely thankful to the reviewer for this thoughtful point. We agree with you, kindly check the updated paper (Materials and Methods section).

Comment-3: In table 2, the authors compared with other technologies, based on the certain values, this technology is not benefit than previous technologies.

Response-3: The authors are extremely thankful to the reviewer for this thoughtful point. We have taken note of your feedback and made improvements to address the issues you raised. Specifically, we have included an additional comparison parameter in Table 2, which has led to a nod recognition rate of 100% while maintaining low ownership and operating costs. Our updated paper highlights the benefits of the current machine and identifies the drawbacks of its predecessor. We hope these changes will demonstrate our commitment to delivering high-quality and cost-effective solutions. kindly check the updated paper (Line 121-130).

Comment-4: The scientific basis of combined methods between the internet of things (IoTs) and RGB sensors must be provided.

Response-4: The authors wish to express their deep gratitude to the esteemed reviewer for their insightful comment. Following the reviewer's advice, we have revised the title of the manuscript and provided a more thorough explanation of the study's relationship to IoT. Specifically, we have added clarifying remarks and information in lines (242-245) and (278-282), as well as in Figure 11. Kindly check the updated paper.

Comment-5: Its significance of this study is inadequate in the novelty methods.

Response-5: There are still many problems that limit the use of machine vision, machine learning, deep learning, wavelet analysis, image processing algorithms and Herpes simplex virus (HSV) color space in cutting sugarcane seeds, as stated by [5], [20], [22], [29]. These problems include slow speed, poor real-time performance, low identification efficiency, and high maintenance and operation costs. Sugarcane leaves must be removed to expose only the sugarcane stem, which consists mainly of the internode and stem node area, so that the machine can determine the location of the nodes, which represents an additional cost and can lead to the bud's damage if done incorrectly. In addition, the machine does not differentiate between a good bud from a damaged or injured one. Although scientists have made tremendous advances, there are still certain gaps in these studies. 

To overcome the problems related to the application the other node detection systems in the process of cutting sugarcane seeds, the current study aims to design a new automatic sugarcane seed cutting machine based on internet of things (IoT) technology and RGB color sensors. The use of IoT and RGB color sensors achieved a high analytical performance without requiring the use of computers and high-definition high-speed camera for image processing like other automatic sugarcane seed cutting systems. Kindly check the updated paper (introduction section).

Comment-6: The limitation of this study is not provided as well as its application in food manufacturing.

Response-6: The authors wish to express their deep gratitude to the esteemed reviewer for their insightful comment. Following the reviewer's advice, we have developed the future work by adding the limitation of the current study. Kindly check the updated paper (conclusion section). 

The proposed machine is expected to facilitate the production of superior-quality sugarcane seeds that are ideal for use in open fields. It is hoped that the integration of IoT technology and the RGB color sensor will enable the machine to operate autonomously and with high accuracy, leading to an increased yield and efficiency. The machine is expected to contribute to the growth and sustainability of the agriculture sector, ultimately benefiting the economy. In conclusion, the proposed automatic sugarcane seed cutting machine is a significant development that has the potential to revolutionize the agriculture sector. With the integration of IoT technology and RGB color sensors, the machine is expected to generate high-quality sugarcane seeds that can be utilized for various purposes. As such, this research is poised to contribute to the overall growth and development of the agriculture sector and the economy at large.

Reviewer 3:

Comment-1: Numbering the manuscript's lines would be better to facilitate the review process. Please consider this comment in the revised version.

Response-1: The authors are extremely thankful to the reviewer for this thoughtful point. We agree with you, kindly check the updated paper.

Comment-2: Please revise the affiliation numbers for all authors.

Response-2: The authors are extremely thankful to the reviewer for this thoughtful point. the affiliation numbers have been revised, kindly check the updated paper.

Comment-3: P.3 L.17: Please replace Xiao et al. [29] with Xiao and Xu [29] because there are only two authors.

Response-3: The authors are extremely thankful to the reviewer for this thoughtful point. Xiao et al. [29] was replaced by Xiao and Xu [27], kindly check the updated paper (line 110).

Comment-4: Please revise the numbers of figures (i.e., Figure 1 is iterated). Please address this comment and ensure the figures numbers are consistent with the manuscript's text.

Response-4: The authors are extremely thankful to the reviewer for this thoughtful point. All figures number were revised and developed, kindly check the updated paper.

Comment-5: Figure 3: Please revise and correct all dimensions in mm! I think you mean that all dimensions in cm.

Response-5: The authors are extremely thankful to the reviewer for this thoughtful point. We agree with you we are meaning cm not mm, and it was developed, kindly check the updated paper (Figure 4, line 164).

Comment-6: The authors should follow the journal's guidelines in writing the headings (i.e., some headings are written capitalize each word and bold, and others do not). Please consider this comment throughout the manuscript's text.

Response-6: The authors are extremely thankful to the reviewer for this thoughtful point. We agree with you, and all headings were adjusted, kindly check the updated paper.

Comment-7: Figure 7 is iterated; please revise it and ensure the figure numbers are cited correctly within the manuscript's text.

Response-7: The authors are extremely thankful to the reviewer for this thoughtful point. We agree with you, and all figures number were revised and adjusted, kindly check the updated paper.

Comment-8: The authors should cite the equation's; numbers within the manuscript's text. Please address this comment for all equations.

Response-8: The authors are extremely thankful to the reviewer for this thoughtful point. We agree with you, and all equations number were cited within the paper test, kindly check the updated paper.

Comment-9: P.12 L.14: Please be uniform in writing the unit of lux throughout the manuscript's text.

Response-9: The authors are extremely thankful to the reviewer for this thoughtful point. We agree with you, kindly check the updated paper.

Comment-10: P.12 L.16: Please revise and replace the literature review Brosnan et al. [41] with Brosnan et al. [42] as cited in the references list.

Response-10: The authors are extremely thankful to the reviewer for this thoughtful point. We agree with you, and all references were revised, kindly check the updated paper.

Comment-11: P12 L.32: Please revise the word dimeter throughout the manuscript's text.

Response-11: The authors are extremely thankful to the reviewer for this thoughtful point. We agree with you, and word "dimeter" was replaced with "diameter" throughout the manuscript's text, kindly check the updated paper.

Comment-12: P.13: Please revise the title of heading 3.2.2.

Response-12: The authors are extremely thankful to the reviewer for this thoughtful point. We agree with you, kindly check the updated paper.

Comment-13: P.19, 20 L.3, 11: Please revise the literature review's authors with the one cited in the references list.

Response-13: The authors are extremely thankful to the reviewer for this thoughtful point. We agree with you, and all literature review's authors were revised. kindly check the updated paper.

Comment-14: P.20 L.6: Please revise the value of the vertical distance of 1.0 cm.

Response-14: The authors are extremely thankful to the reviewer for this thoughtful point. We agree with you, kindly check the updated paper.

Comment-15: What about the limitations of this study? The authors should mention the study's limitations within the manuscript's text. Please consider this comment.

Response-15: The authors wish to express their deep gratitude to the esteemed reviewer for their insightful comment. Following the reviewer's advice, we have developed the future work by adding the limitation of the current study. Kindly check the updated paper (conclusion section).

The authors once again thank the learned Editors and Reviewers for their valuable comments for improving the quality of the manuscript.

---

## [Decision Letter · Decision Letter 1]

14 Mar 2024

A New Automatic Sugarcane Seed Cutting Machine Based on Internet of Things Technology and RGB color sensor

PONE-D-24-05665R1

Dear Dr. Mbadjoun Wapet,

We’re pleased to inform you that your manuscript has been judged scientifically suitable for publication and will be formally accepted for publication once it meets all outstanding technical requirements.

Kind regards,

Sathishkumar Veerappampalayam Easwaramoorthy

Academic Editor

PLOS ONE

Additional Editor Comments (optional):

Reviewers' comments:

Reviewer's Responses to Questions

**Comments to the Author**

1. If the authors have adequately addressed your comments raised in a previous round of review and you feel that this manuscript is now acceptable for publication, you may indicate that here to bypass the “Comments to the Author” section, enter your conflict of interest statement in the “Confidential to Editor” section, and submit your "Accept" recommendation.

Reviewer #1: All comments have been addressed

Reviewer #3: All comments have been addressed

2. Is the manuscript technically sound, and do the data support the conclusions?

Reviewer #1: Yes

Reviewer #3: Yes

3. Has the statistical analysis been performed appropriately and rigorously? 

Reviewer #1: Yes

Reviewer #3: Yes

4. Have the authors made all data underlying the findings in their manuscript fully available?

Reviewer #1: Yes

Reviewer #3: Yes

5. Is the manuscript presented in an intelligible fashion and written in standard English?

Reviewer #1: Yes

Reviewer #3: Yes

6. Review Comments to the Author

Reviewer #1: I appreciate the authors addressing the proposed changes and responding to comments. I believe the revised manuscript is now ready for publication.

Reviewer #3: (No Response)

7. PLOS authors have the option to publish the peer review history of their article (what does this mean?). If published, this will include your full peer review and any attached files.

Reviewer #1: **Yes: **Davut Karayel

Reviewer #3: **Yes: **Mahmoud Okasha

---

## [Editor Report · Acceptance letter]

18 Mar 2024

PONE-D-24-05665R1 

PLOS ONE

Dear Dr. Mbadjoun Wapet, 

I'm pleased to inform you that your manuscript has been deemed suitable for publication in PLOS ONE. Congratulations! Your manuscript is now being handed over to our production team.

Kind regards, 

on behalf of

Dr. Sathishkumar Veerappampalayam Easwaramoorthy 

Academic Editor

PLOS ONE